# Characteristics of an Eight-Quadrant Corner Reflector Involving a Reconfigurable Active Metasurface

**DOI:** 10.3390/s22134715

**Published:** 2022-06-22

**Authors:** Lin Gan, Guang Sun, Dejun Feng, Jianbing Li

**Affiliations:** State Key Laboratory of Complex Electromagnetic Environmental Effects on Electronics and Information System, National University of Defense Technology, Changsha 410073, China; ganlin16@nudt.edu.cn (L.G.); sunguang17@nudt.edu.cn (G.S.); fdj117@163.com (D.F.)

**Keywords:** active frequency-selective surface, corner reflector, electromagnetic scattering characteristic, miniaturization

## Abstract

The traditional corner reflector is a type of classical passive jamming equipment but with several shortcomings, such as fixed electromagnetic characteristics and a poor response to radar polarization. In this paper, an eight-quadrant corner reflector equipped with an electronically controlled miniaturized active frequency-selective surface (MAFSS) for X band is proposed to obtain better radar characteristics controllability and polarization adaptability. The scattering characteristics of the new eight-quadrant corner reflector for different switchable scattering states (penetration/reflection), frequency and polarization are simulated and analyzed. Results show that the RCS modulation depth, which is jointly affected by the electromagnetic wave frequency and incident directions, can be maintained above 10 dB in the majority of directions, and even larger than 30 dB at the resonant frequency. Moreover, the RCS adjustable bandwidth can be as wide as 1 GHz in different incident directions.

## 1. Introduction

As a classic passive jamming device, the corner reflector can obviously interfere with and degrade radars due to the strong backward radar cross-section (RCS) [1,2], while it also has obvious defects, such as fixed characteristics, limited bandwidth and being sensitive to incident wave polarization. Some researchers have designed dihedral and trihedral corner reflectors with variable structures containing disassembly, rotation and other structures to change the RCS [3,4], but structurally variable corner reflectors require manual or mechanical control of metal plates, making it difficult to realize the interpulse modulation. Moreover, a change in mechanical structures is unable to solve the problems of limited bandwidth and polarization sensitivity.

Over the past several years, increasing attention has been paid to the frequency-selective surfaces (FSS) due to their filter characteristics and low complexity in design and fabrication [5], which is gradually developing towards broadband [6], multi-frequency [7,8] and high angular stability [9,10]. However, the traditional FSS can only be used in a specific frequency band after manufacturing, so the reconfigurable FSS that can better adapt to the complex electromagnetic environment have been rapidly developed. By loading active devices such as PIN diodes on the FSS, effects including filtering characteristics adjustment and mode switching can be obtained [11,12,13,14].

In the communication field, such as point-to-point wireless links, many researchers have combined FSS with corner reflector antennas to select frequency bands [15], reconstruct beams [16,17,18] and improve antenna gains [19,20,21], so as to reduce multipath effects and improve the performance of communication links. In addition, corner reflectors equipped with FSS can also be used in the wireless local positioning system [22,23] and indoor localization [24]. If electronically controlled angle reflectors involving FSS are applied in the radar interference field, not only the above defects of traditional corner reflectors could be improved, but also the pulse-to-pulse phase and amplitude modulation of radar wave could be possible, resulting in various complex interference effects on radar detection, imaging, etc.

In practice, the metasurface is composed of many small unit cells. These may generally result in a grating lobe effect when the incident wave is oblique to the surface. To solve this problem, the miniaturization of the unit cell is recommended [25,26,27,28,29,30]. Moreover, the small size makes the unit cell less sensitive to the incident polarization, and the penetration effect can also be improved by increasing the number of unit cells within the limited space. These all coincide with the design requirements of the corner reflector.

In this paper, an electronically controlled eight-quadrant corner reflector equipped with a miniaturized wave-penetrating active frequency-selective surface is proposed, and the scattering characteristics are analyzed by CST Microwave Studio. Compared with dihedral and trihedral corner reflectors, the eight-quadrant corner reflector can greatly increase the effective angle width and intensity of scattering, which can better adapt to various application scenarios. By changing the bias voltage loaded on the metasurface to change the reflectivity, the RCS and scattering states of the eight-quadrant corner reflector can be dynamically adjusted, allowing it to adapt to different interference requirements in various applications.

## 2. Proposed Miniaturized Active Frequency-Selective Surface

A preliminary design of the unit cell for the X band is presented here. It consists of two metallic layers, one dielectric layer and two variable PIN diodes. As illustrated in Figure 1, it is rotationally symmetric and all the patterns are printed on a F4BM substrate with a thickness of 0.5 mm and a relative permittivity of 2.2. The thickness of the metallic layer is 0.005 mm. The resonant wavelength is increased by bending the metal plating to realize the miniaturization. The loaded variable PIN diodes can not only increase the capacitance and inductance value in the equivalent resonant circuit, but also make the electromagnetic characteristics adjustable. Moreover, the upper and lower metal layers are symmetrical and connected with each other by metallic pillars, which can improve the stability to the polarization information.

In practical applications, the resistance of the PIN diode can be changed by adjusting the bias voltage. The lumped RLC boundary condition is used to replace the PIN diode in our simulation, and the variation of the PIN diode bias voltage can be represented by changing the resistance value. Reflection coefficients of the unit cell at normal incidence are shown in Figure 2 with respect to horizontal/vertical polarizations. It is observed that for both the horizontal and vertical polarizations, the reflection coefficients show significant differences along with the resistances of PIN diodes, but they all show the same trend of first decreasing and then increasing. The similarity of Figure 2a,b well indicates the insensitivity of the unit cell to incident polarization. The curves under the horizontal and vertical polarization in the X band reach the lowest and highest, respectively, when the resistance values are 1.5 Ω and 5000 Ω, which namely represent the wave-penetrating state and the reflection state. When the resistance is 1.5 Ω, reflection coefficients are less than −10dB from 10.3 GHz to 12 GHz. The resonant frequencies are both around 11.87 GHz for the two polarization cases, and the corresponding reflection coefficients can reach −27.46 dB and −39.86 dB, respectively. Of course, when the size parameters change, the penetrating frequency may change accordingly. If we wish to have the cell penetrating at a concerned frequency in the X band, it is possible to design the size and other parameters.

Figure 3 presents the reflection coefficients for oblique incidence when the resistance is 1.5 Ω. It can be found from the curves that the unit cell can maintain stable penetration characteristics between 0° (normal incident) and 60°, except for slight shifts in the resonant frequency.

Based on the above characteristics, it can be concluded that the unit cell has good angle and polarization stability. In this paper, we propose an eight-quadrant corner reflector with the middle metal plate replaced by a metasurface composed of the above unit cells. The RCS of this reflector can be dynamically changed by adjusting the bias voltage, which makes it possible to simulate targets of different RCS, and obtain rapid modulation effects between and within signal pulses.

## 3. Electromagnetic Scattering Characteristics of the Proposed Eight-Quadrant Corner Reflector

The eight-quadrant corner reflector is a structural extension of the trihedral corner reflector, which can strongly backscatter incident electromagnetic waves in a much larger solid angle domain. The overall structure is shown in Figure 4, with θ and ϕ being the elevation and azimuth angle, respectively. Based on the traditional corner reflector, the electronically controlled corner reflector in this paper only replaces the middle metal plate with a wave-penetrating active frequency selection surface, which appears black in Figure 4. The metasurface is composed of 30 × 30 unit cells, shown in Figure 1, and the corner reflector size is 132 mm × 132 mm × 132 mm. Since the eight quadrant structures are exactly the same, and the wave-transmitting unit shown in Figure 1 has a similar change trend under the horizontal and vertical polarization conditions of the incident wave, this paper only presents the simulation results for one of the quadrants under the horizontally polarized incidence.

### 3.1. RCS Modulation Depth

The RCS difference between the reflection and penetration state is defined as the modulation depth Δσ:(1)$$\Delta \sigma =10\mathrm{lg}({\mathrm{RCS}}_{\mathrm{ref}}/{\mathrm{RCS}}_{\mathrm{pen}})$$
where ${\mathrm{RCS}}_{\mathrm{ref}}$ and ${\mathrm{RCS}}_{\mathrm{pen}}$ represent the RCS of the corner reflector in the reflection and penetration state, respectively. Thus, the greater Δσ means the better electromagnetic switchability of the corner reflector. The RCS patterns and modulation depth at several typical X-band frequency points under different directions and resistances are illustrated in Figure 5 and Figure 6. It can be seen that the electronically controlled corner reflector exhibits a strong scattering state when the resistance is 5000 Ω. Modulation depth at these three frequencies (10 GHz, 11 GHz, 12 GHz) shows significant differences as the incident direction changes. At 10 GHz, Δσ is maintained between 0 and 6 dB when θ is 0∼60°, which is relatively lower. At 11 GHz and 12 GHz, high RCS fluctuations can be maintained in a larger incident range. While *f* = 11 GHz, Δσ is maintained above 10 dB for different azimuth directions when θ is 0∼67°, except the areas enclosed by two 10 dB dashed lines, and it can even reach up to 26.26 dB. At 12 GHz, Δσ in different azimuth directions exceeds 10 dB when θ is 0∼13° and 41∼60°, which is slightly lower than that of 11 GHz, and can reach up to 26.31 dB at θ = 51°.

In addition, Δσ under the two resistance values is not strictly symmetrical about ϕ = 45°. The main reason is that the incident waves from 0∼45° and 45∼90° are symmetrical with respect to metallic layers due to the rotationally symmetric structure of metasurface unit cells, but the angle between PIN diodes and the incident wave is always changing, leading to a certain asymmetry of the total RCS about the azimuth angle ϕ. Figure 2 indicates that the resonant frequency is located at 11.87 GHz, where the metasurface has the greatest degree of wave penetration. Therefore, Δσ can also reach the peak at 11.87 GHz theoretically. Simulation results in Figure 7 show that the pattern at the resonant frequency is similar to that of 12 GHz, but Δσ are significantly higher, which can reach up to 33.58 dB at ϕ = 19°, θ = 51°.

Based on the above simulation results, it can be concluded that Δσ is jointly affected by the incident frequency and direction, and the RCS modulation effect is relatively reduced when θ exceeds approximately 60°. This is because the wave penetration effect may deteriorate when the incident angle to the metasurface is too small. This phenomenon can be improved by replacing other metal plates with metasurfaces as well, but the cost of this reflector will increase accordingly.

### 3.2. Adjustable Bandwidth

For the electronically controlled corner reflector, maintaining large RCS fluctuations over a wider bandwidth indicates better frequency adaptation. The above simulation results only present Δσ at several discrete frequency points, from which the bandwidth with good modulation effect is not available. To investigate the bandwidth, the change in Δσ along with frequency in different incident directions is presented in Figure 8.

It can be seen from the subfigures that the resonant frequency shifts slightly as the incident directions change, but all lie between 10.5 GHz and 12 GHz, and the adjustable bandwidth corresponding to 10 dB and above in different incident directions is larger than 1 GHz. In particular, the electronically controlled corner reflector can provide a wider adjustment bandwidth and a larger wave penetration depth at θ = 60°.

### 3.3. Comparison with the Traditional Eight-Quadrant Corner Reflector

Due to the equipped metasurface, the scattering characteristics of the electronically controlled corner reflector are inevitably different from those of the traditional one. Figure 9 presents the RCS curves of the two corner reflectors with the same size in different incident directions and at several frequencies in the X band.

When the resistance is 5000 Ω, it can be seen from Figure 9a,b that the overall trends of the two corner reflectors’ RCS against frequency and elevation angle are similar, but the latter curves are slightly deformed. The RCS of the electronically controlled corner reflector is smaller, which shows a significant downward trend, especially in the case of ϕ = 60° in Figure 9b. This indicates that full reflection of the electromagnetic wave cannot be realized due to a certain energy loss, even if the metasurface is in a reflective state. Moreover, the RCS of the electrically controlled corner reflector with 1.5 Ω is significantly lower compared to the other two, especially between 0° and 60°, and grows significantly when θ exceeds 60°.

In general, although the electronically controlled corner reflector has a certain degree of energy loss in the strong reflection state, it can maintain the original scattering efficiency of the traditional corner reflector well with 5000 Ω resistance and has the ability of deep RCS modulation with 1.5 Ω resistance.

## 4. Conclusions

In this paper, an electronically controlled eight-quadrant corner reflector loaded with a wave-transmitting active frequency-selective surface is studied. Simulation results show that this reflector has the advantage of rapid RCS modulation capability in the wide solid angle domain. The modulation depth can reach 33.58 dB at the resonant frequency, and the adjustable bandwidth corresponding to 10 dB and above can exceed 1 GHz. The parameters of the metasurface will be further optimized to enable better modulation capability over a wider range within the specific frequency band of concern. The good electromagnetic switchability of the electronically controlled corner reflector makes it suitable for typical target simulation, radar interference and other fields.

## Figures and Tables

**Figure 1 sensors-22-04715-f001:**
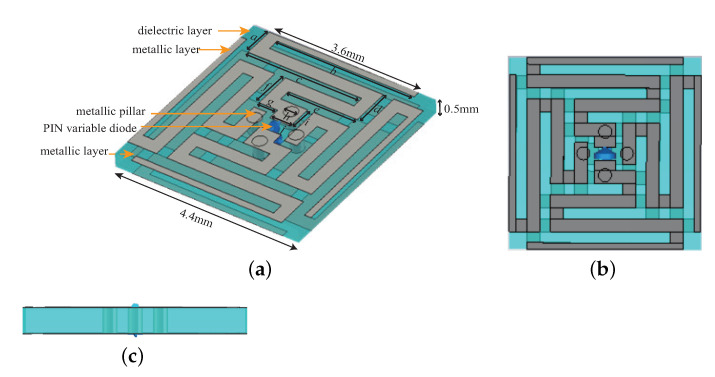
Three−dimensional schematic diagram of the unit cell (dimensions: *a* = 0.7 mm, *b* = 3.3 mm, *c* = 2.75 mm, *d* = 0.85 mm, *e* = 1.65 mm, *f* = 0.85 mm, *g* = 0.45 mm, *h* = 0.3 mm, *i* = 0.55 mm, *j* = 0.5 mm. (**a**) Oblique view. (**b**) Top view. (**c**) Side view.

**Figure 2 sensors-22-04715-f002:**
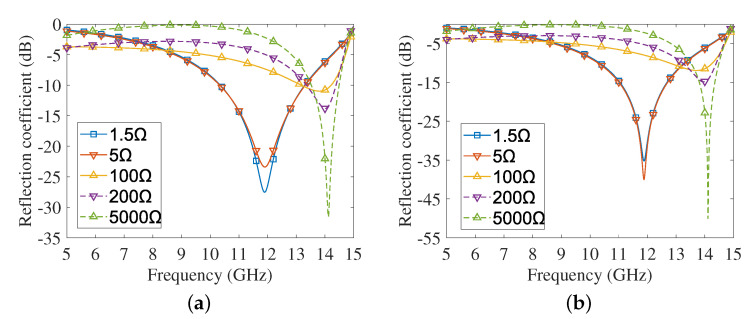
Normal incident reflection coefficient against frequency (5∼15 GHz). (**a**) Horizontal polarization. (**b**) Vertical polarization.

**Figure 3 sensors-22-04715-f003:**
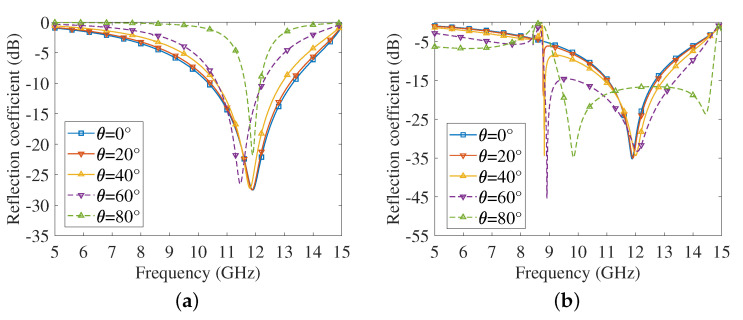
Oblique incident reflection coefficient against frequency (5∼15 GHz). (**a**) Horizontal polarization. (**b**) Vertical polarization.

**Figure 4 sensors-22-04715-f004:**
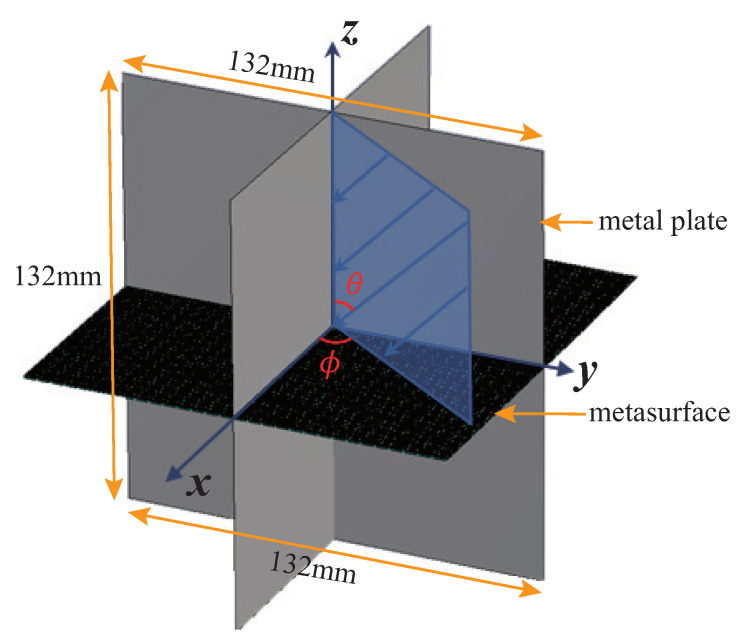
The structure of the electronically controlled eight-quadrant corner reflector.

**Figure 5 sensors-22-04715-f005:**
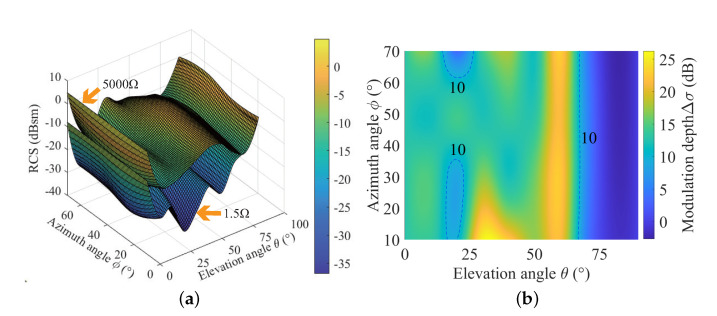
The RCS pattern and modulation depth at 11 GHz. (**a**) The RCS pattern. (**b**) RCS modulation depth.

**Figure 6 sensors-22-04715-f006:**
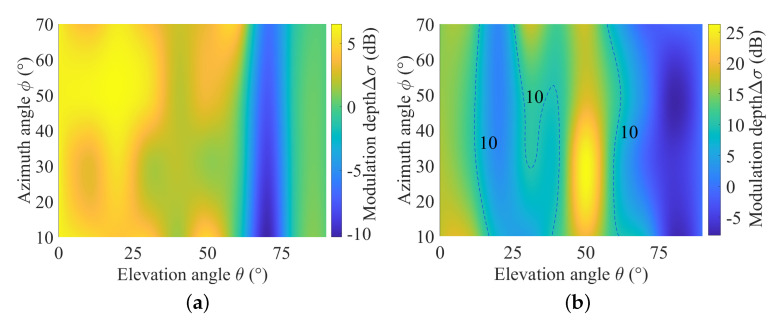
The RCS pattern and modulation depth at 10 GHz and 12 GHz. (**a**) 10 GHz. (**b**) 12 GHz.

**Figure 7 sensors-22-04715-f007:**
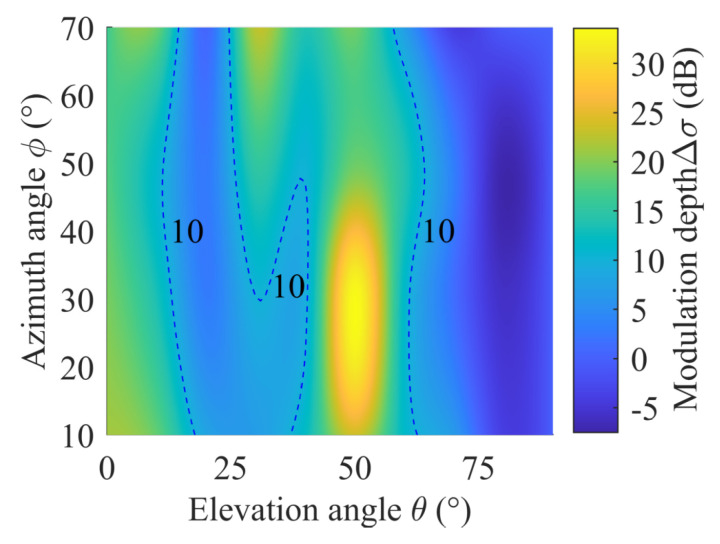
RCS modulation depth at the resonant frequency.

**Figure 8 sensors-22-04715-f008:**
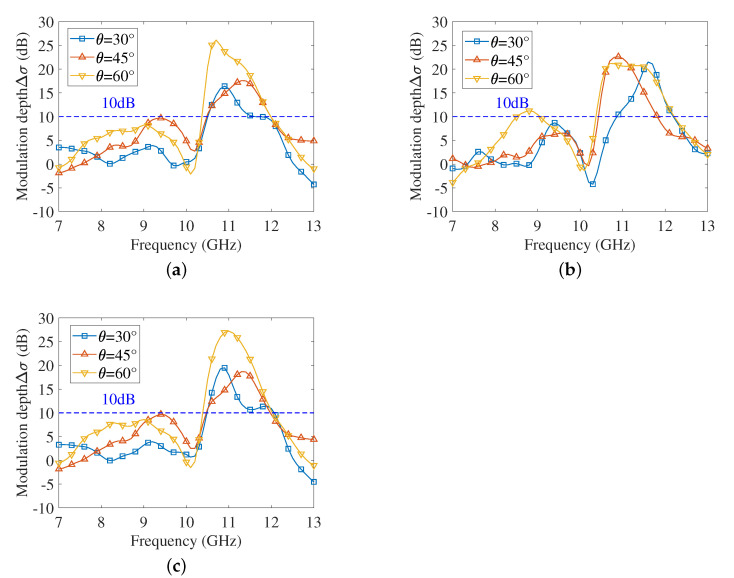
RCS modulation depth in different incident directions. (**a**) ϕ = 30°. (**b**) ϕ = 45°. (**c**) ϕ = 60°.

**Figure 9 sensors-22-04715-f009:**
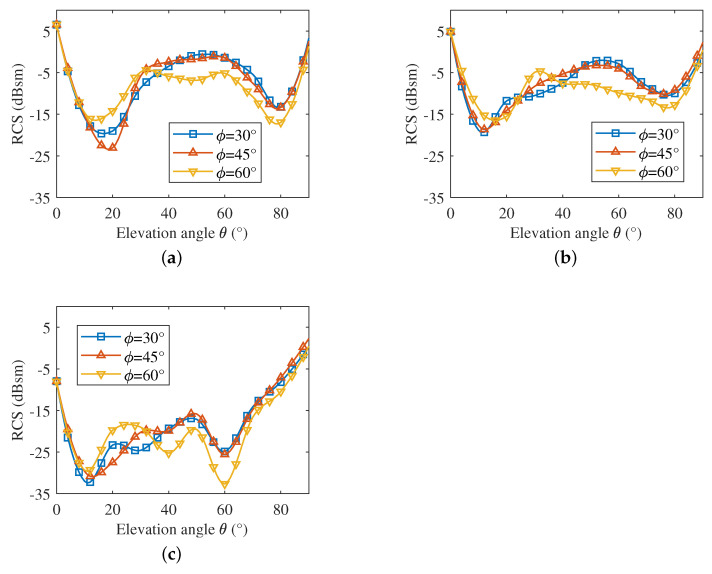
RCS comparison of two corner reflectors at 11 GHz. (**a**) Traditional corner reflector. (**b**) Electrically controlled corner reflector with 5000 Ω resistance. (**c**) Electrically controlled corner reflector with 1.5 Ω resistance.

## Data Availability

Not applicable.

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
