# Peer review of "Characteristics of an Eight-Quadrant Corner Reflector Involving a Reconfigurable Active Metasurface"

_sensors, 2022, doi:10.3390/s22134715_

Round 1

Reviewer 1 Report

This communication appears interesting that can control RCS from corner reflectors ultimately with reconfigurability. This part of the paper is interesting and I think the title could better reflect the work to be:
"Characteristics of an Eight-Quadrant Corner Reflector Involving a Reconfigurable Active Metasurface"

If I understand right the PIN diodes on the unit cells are controlling the impedance. It would be useful to appreciate how this is done since surely some switched states give fixed impedances? Thus a table may be useful. Also is there actually a need for reconfigurability? The benefit would be that we could adjust the RCS to high and low to create some spoof reflectors that make a more effective decoy to radar. This would be worth noting for the reconfigurability benefit and what degree of control is possible.

It is hard to tell from Figure 4 if the metasurface is only shown in one quadrant for clarity as surely would it not be necessary in others too? A note in the caption and text would be usfeul.

For figures 2-3 it would be better to have the max value in all graphs as 0dB since the reflection coefficient cannot get higher and we can then see how high the reflection is in some impedances.

For RCS modulation depth it would be good to refer to a suitable reference that defines/derives it fully. Adding an equation from the reference wouldn't harm either so then the reader can readily see and interpret the result. It states that we are taking the ratio from reflection to penetration but where is the penetration going into since we have a conducting surface and metasurface?

Spacing between parentheses such as where there is surfaces(FSS) and RCS[3,4] must be present

Please do proof check the paper as some typographical errors like the following are shown:
"..increasing attention HAS been paid..."
"..small size makeS the unit cell.."

Reviewer 2 Report

To improve the characteristics controllability and polarization adaptability of corner reflector, this manuscript proposed an eight-quadrant corner reflector with the miniaturized active frequency selective surface (MAFSS) in X band. In this passive jamming equipment, it’s RCS and scattering states can be dynamically adjusted for different application scenarios. The adaptability of the corner reflector is verified by CST simulation data. In my opinion, the theoretical and the details of the presented corner should be described in more detail. In addition, the jamming equipment and corner should also be compared (in performance) with the existing metasurface corner reflector. Several points should be mitigated previous to publication. Detailed comments are provided below.

1. The review of existing studies of the metasurface is insufficient, and more references to recent metasurface studies should be considered

2. What is the theoretical model of the design of the metasurface and the corner reflector?

3. Can you provide a more detailed metasurface design by providing different perspectives?  What’s more, the thickness of the metallic layer is necessary for the metasurface.

4. Can you analyze the cause of slight shifts of the resonant frequency which is shown in Fig. 3 (b)?

5. In addition to traditional angular reflection, are there related angular reflection studies using metasurface that can be compared to verify performance?

6. The resolution of Fig. 8 is different from other figures, it is better to give a clear version.

Round 2

Reviewer 1 Report

The paper revision is very good. I only have one very minor correction to change 'lg' to 'log in equation (1) and all should be fine.

Reviewer 2 Report

The author has basically modified the manuscript according to my comments, I think the manuscript can be accept for publication in this form.

Author Response

Thanks very much for your valuable help. We have removed the previous reference [22] due to the little relevance to our paper.